# Do the shuffle: Exploring reasons for music listening through shuffled play

**Katie Rose M. Sanfilippo**[1], **Neta Spiro**[2,3]\*, **Miguel Molina-Solana**[3,4], **Alexandra Lamont**[5]

**1** Psychology Department, Goldsmiths, University of London, London, United Kingdom, **2** Centre for Performance Science, Royal College of Music, London, United Kingdom, **3** Imperial College London, London, United Kingdom, **4** University of Granada, Granada, Spain, **5** School of Psychology, Keele University, Keele, United Kingdom

\* neta.spiro@rcm.ac.uk

**Data Availability Statement:** All of the raw data is freely available and can be downloaded for reuse from [https://miguems.github.io/dotheshuffle/], a

## Abstract

Adults listen to music for an average of 18 hours a week (with some people reaching more than double that). With rapidly changing technology, music collections have become overwhelmingly digital ushering in changes in listening habits, especially when it comes to listening on personal devices. By using interactive visualizations, descriptive analysis and thematic analysis, this project aims to explore why people download and listen to music and which aspects of the music listening experience are prioritized when people talk about tracks on their device. Using a newly developed data collection method, *Shuffled Play*, 397 participants answered open-ended and closed research questions through a short online questionnaire after shuffling their music library and playing two pieces as prompts for reflections. The findings of this study highlight that when talking about tracks on their personal devices, people prioritise characterizing them using sound and musical features and associating them with the informational context around them (artist, album, and genre) over their emotional responses to them. The results also highlight that people listen to and download music because they like it–a straightforward but important observation that is sometimes glossed over in previous research. These findings have implications for future work in understanding music, its uses and its functions in peoples' everyday lives.

## Introduction

Music listening is a ubiquitous and constant phenomenon in industrialised society. Adults listen to music for an average of 18 hours a week (with some participants reaching more than double that) [1]. Music is heard between 44% and 68% of people's waking hours, accompanying a range of activities such as travel, eating, exercise, work and study [2–4]. While music is found in many public settings including transport, shops, restaurants and the workplace, self-chosen individual music listening represents an important aspect of musical engagement. North et al. [3] reported 37% of music listening episodes were undertaken alone, and of these, 82% were episodes of self-chosen music. Similarly, Greasley and Lamont [1] found 55% of music listening experiences were when participants were alone. Emotional reactions to music

webpage specifically designed to accompany this paper.

**Funding:** This work is funded by the EU H2020 programme (ga No. 743623). NS has been supported by HEartS, a project funded by the UK's Arts and Humanities Research Council to investigate the health, economic and social impact of the arts (grant ref. AH/P005888/1) to MMS. The funder had no role in study design, data collection and analysis, decision to publish, or preparation of the manuscript.

**Competing interests:** The authors have declared that no competing interests exist.

have been shown to be stronger when music is chosen by its listeners [4], and the increase in availability and portability of personal listening devices suggests that people are now able to choose music and curate music libraries in an increasing range of situations and for a range of activities.

Building on research from experimental aesthetics investigating the fit between music and context, recent research has begun to explore the role of situation. For instance, North and Hargreaves [5] found that listeners preferred high arousal versions of a piece while exercising and low arousal versions of the same piece while relaxing. More recently, researchers have investigated both the contexts and the activities that are associated with music, finding differences in both [1,3,6]. For instance, music in the gym is found to be more motivating than music in a restaurant [6]. Greb, Schlotz and Steffens [7] directly compared the effects of individual and situational influences on different functions of music listening, asking participants to imagine and then describe self-chosen listening situations. They found situational influences to be more important (see also [8]), concluding that active engagement with music can only be understood when accounting for context.

Similarly, in terms of functions of music listening, research has identified a range of motivations for listening [9] and functions that music serves [10]. A key motivation for (and outcome of) music listening is its emotional response [9,11–13]. Other motivations include social relationships and connections, seeking information about music, and music's potential for movement (tapping along or dancing) [9,14]. Music serves a range of functions including intellectual stimulation, coping with problems, motor synchronisation and wellbeing, overcoming loneliness, distraction, and defining and marking personal and social identity [7,13,14]. The pleasure of music listening has been found to result from the motives of relaxation, power, joy, and kinship [15].

Functions and uses have also been linked to aspects of the music and music preference. In an experimental study Sallavanti, Szilagyi and Crawley [16] found listeners used more complex music for cognitive purposes and less complex music for emotional purposes. Schäfer and Sedlmeier [17] found people who had strong preferences for their favourite music, whatever it might be, also emphasised the communicative functions of music listening such as expressing one's own identity (cf. [18]), and tended to show more functions of their listening behaviour (cf. [1]). While mood, arousal and emotional benefits were common across all styles, they also found that fans of different styles emphasised different functions of their own preferred music. For instance, electronic music fans appreciated the sense of energy and ability of the music to put them in a good or ecstatic mood, along with understanding one's own thoughts and feelings, while pop music fans liked the music's ability to express their values and the sense of identification with artists and other people.

Much of the research reviewed so far has either presented participants with experimenter-selected music or asked people to report preferred styles, recreate memories of their favourite music, or talk about the music they like. One main drawback of these approaches is that participants either have no choice of musical stimulus or they are relying on memory which, although sometimes vivid [19,20], is not always accurate [21]. A more robust way of exploring preference is to focus on people's own music collections. In studies that have done this, participants have been invited to select, discuss, and even play their favourite music during an interview [22]. However, even in an interview study that does this (e.g. Greasley, Lamont and Sloboda [22]) participants' favourite music was emphasised, thus excluding their perceptions of music in their collection that may not be their favourite or even music they like.

Rapidly changing technology means the physical collection as studied by Greasley, Lamont, and Sloboda [22] in the mid-2010s has become overwhelmingly digital. Krause, North, and Hewitt [23] found that young adults now carry most of their music with them on personal

listening devices; and thus data from playlists is beginning to be deliberately used in research. For instance, Krause and North [24] asked participants to create playlists for specific situations, confirming earlier findings that music choices can be heavily influenced by situation. This reflects a more curated approach to research than experience sampling, which aims to capture random slices of listening experiences. Between these two poles, random or 'shuffle' modes of listening provide an interesting midpoint between control and chance. Shuffle provides a more flexible approach to musical engagement [24], and is a very popular listening strategy, as part of the new flexibility that technology affords [25]. For instance, 35% of the mobile participants in Heye and Lamont's [26] study were listening via shuffle mode (the largest response category) while travelling.

### Aims and research questions

The current study bridges the gap between reflective interviews about curated self-chosen music and random experience sampling of music listening in everyday settings. The aim of this project is to investigate which aspects of the music listening experience are prioritized by people listening to music on their personal devices. We do so by developing a new data collection method, *Shuffled Play*. This method does not specifically focus on people's favourite music or music played in the moment. Instead it starts from a quasi-random selection of music that people have chosen as part of their personal music libraries which they access using the shuffle function. With this method, along with collecting mixed response type data, we build on and go beyond the more typical foci in the literature on music and emotion, music in everyday life and music and technology.

In this study we specifically explore two main research questions through a short online questionnaire asking participants to shuffle their music library and play two tracks as a prompt for reflection:

1. Why do people download and listen to music?

2. Which aspects of the music listening experience are prioritized when people talk about a track on their device?

## Materials and methods

### Design

For this descriptive study, we collected responses from a mix of question response types, including closed and open questions. We administered a short online questionnaire asking participants to shuffle their music library on their listening device and answer open and closed questions about the two music tracks that were selected first. Ethical approval was obtained from the Nordoff Robbins Ethics Committee and Keele University. All written consent was obtained online through the online questionnaire.

### Participants

398 people began the survey and 397 completed information regarding music on their device. Responses from these 397 participants were used in the analysis of participants' open-ended responses to their music. Demographic information was completed by 322 participants. About two thirds of the 322 participants identified as female ($n$ = 202, 63%) and all were between the ages of 18 and 66 years old ($M$ = 30.15, $SD$ = 11.36). Sixty-nine percent of the participants were aged between 18 and 35 ($n$ = 223, 69%). The majority were from, or currently resided in, the UK ($n$ = 226, 83% and $n$ = 257, 79.5% respectively) (Table 1).

**Table 1. Demographic characteristics of the 322 participants.**

| | n (%) out of 322 |
|---|---|
| **How do you describe yourself?** | |
| Female | 202 (62.73) |
| Male | 117 (36.33) |
| Would rather not say | 3 (0.93) |
| **How old are you?** | |
| 18–24 | 145 (45.03) |
| 25–34 | 78 (24.22) |
| 35–44 | 56 (17.39) |
| 45–54 | 26 (8.07) |
| 55–64 | 16 (5.00) |
| 65–74 | 1 (0.31) |
| **Which country are you originally from?** | |
| UK | 226 (70.18) |
| Bulgaria | 3 (0.93) |
| Denmark | 1 (0.31) |
| Finland | 1 (0.31) |
| France | 2 (0.62) |
| Germany | 6 (1.86) |
| Greece | 2 (0.62) |
| Ireland | 9 (2.80) |
| Netherlands | 6 (1.86) |
| Poland | 2 (0.62) |
| Portugal | 1 (0.31) |
| Slovakia | 1 (0.31) |
| Slovenia | 1 (0.31) |
| Spain | 2 (0.62) |
| Sweden | 2 (0.62) |
| Switzerland | 1 (0.31) |
| Turkey | 2 (0.62) |
| *Total European countries* | *268 (83.23)* |
| Canada | 5 (1.55) |
| USA | 30 (9.32) |
| *Total North America* | *35 (10.87)* |
| Brazil | 2 (0.62) |
| Brunei | 2 (0.62) |
| China | 1 (0.31) |
| Ghana | 1 (0.31) |
| Kenya | 1 (0.31) |
| New Zealand | 1 (0.31) |
| Nigeria | 2 (0.62) |
| Pakistan | 1 (0.31) |
| Peru | 1 (0.31) |
| Philippines | 2 (0.62) |
| Singapore | 1 (0.31) |
| South Africa | 2 (0.62) |
| Zambia | 1 (0.31) |
| Zimbabwe | 1 (0.31) |

*(Continued)*

**Table 1.** (Continued)

| | n (%) out of 322 |
|---|---|
| *Total other countries* | *19 (5.90)* |
| **In which country do you currently reside?** | |
| UK | 256 (79.50) |
| Belgium | 2 (0.62) |
| Czech Republic | 1 (0.31) |
| Denmark | 1 (0.31) |
| Finland | 1 (0.31) |
| France | 1 (0.31) |
| Germany | 3 (0.93) |
| Greece | 2 (0.62) |
| Ireland | 6 (1.86) |
| Latvia | 1 (0.31) |
| Netherlands | 1 (0.31) |
| Poland | 1 (0.31) |
| Slovenia | 1 (0.31) |
| Spain | 1 (0.31) |
| Sweden | 2 (0.62) |
| *Total European countries* | *280 (86.96)* |
| Canada | 4 (1.24) |
| Mexico | 1 (0.31) |
| United States of America | 30 (9.32) |
| *Total North America* | *35 (10.87)* |
| Algeria | 1 (0.31) |
| Australia | 1 (0.31) |
| Brazil | 1 (0.31) |
| New Zealand | 2 (0.62) |
| Singapore | 1 (0.31) |
| South Africa | 1 (0.31) |
| *Total other countries* | *7 (2.17)* |

Note. Only 70 out of the 322 answered the question "Why do you usually listen to this track?" (described below) due to a glitch in the survey program. No significant differences in demographic information were found between those that answered this question and those that did not.

As Table 2 details, the vast majority of the participants did not use music in their professional life ($n$ = 227, 70%). The average score on the Active Engagement Subscale of the Goldsmiths Music Sophistication Index was 40.50 ($SD$ = 8.19) corresponding to the 43rd percentile of the data norm reported in Müllensiefen et al. [27]. Our sample was thus representative of the average level of musical engagement. The vast majority of participants usually listened to music on their personal listening devices ($n$ = 281, 87%). The two most popular ways of choosing music to listen to were by using self-selected playlists ($n$ = 119, 30%) and choosing particular tracks ($n$ = 146, 45%).

## A new data collection method: Shuffled play

The online questionnaire consisted of three sections. In the first section, participants were asked to use their regular music listening device (e.g. phone, laptop, etc.), turn on the shuffle function within a listening app they usually use (e.g. Spotify, iTunes, etc.) and press play.

**Table 2. Musical experience of the 322 participants.**

|  | n (%) for 322 |
| --- | --- |
| **I use music in my professional life** |  |
| Yes | 95 (29.50) |
| No | 277 (70.50) |
| **Music Sophistication Index: Active Engagement Subscale** |  |
| 10–19 | 2 (0.62) |
| 20–29 | 26 (8.07) |
| 30–39 | 106 (32.92) |
| 40–49 | 141 (43.79) |
| 50–59 | 47 (14.60) |
| **How do you usually listen to your music?** |  |
| On my personal listening devices (through iTunes, Spotify, Soundcloud, etc.) | 281 (87.27) |
| Other | 41 (12.73) |
| **How do you usually choose the music you listen to?** |  |
| Choose which tracks I want to listen to when I want to listen to them | 146 (45.34) |
| Use self-selected playlists | 119 (36.96) |
| Using playlists created by others | 18 (5.59) |
| Radio | 15 (4.66) |
| Other | 24 (7.45) |
| **I usually . . . (select all that apply)** | n (%) for 414 |
| Buy my music online | 116 (28.02) |
| Buy my music in a shop | 11 (2.65) |
| Listen via a subscription to a site (e.g. Spotify) | 194 (46.86) |
| Get my music for free | 81 (19.56) |
| Other | 12 (2.90) |
| *Total* | *414* |

Note. Only 70 out of the 322 answered the question "Why do you usually listen to this track?" (described below) due to a glitch in the survey program. No significant differences in musical experience were found between those that answered this question and those that did not. "I usually. . ." was a select all that apply. Therefore the total % is out of 414 total options that were chosen.

Whatever track came on first, they were asked to provide the track title, artist, device and app they were using. They were then asked open-ended questions about this track: "What's the first thing that comes into your mind about this track?" and "Why did you choose to save this track to your device?". They were asked whether they enjoyed the track on a five point scale from Really enjoy = 5 to Really do not enjoy = 1 followed by an open question: "Why do you enjoy or not enjoy listening to the track?". They were asked about their relationship with the track: "Do you have a relationship with this track? Whether it's 'Yes' or 'No' please explain". The participant then pressed shuffle again and repeated this section about a new track. Once they had responded to all open-ended questions for both tracks they went onto the second section of the questionnaire.

The second section included three multiple-choice questions about the tracks they had just written about. Participants were presented with all the closed questions to be answered in relation to the first track they discussed in the previous section. They were reminded of the track title and artist in the questionnaire. They then answered all the questions about the second track they had discussed in the previous section, again being reminded of the track's title and artist. The first, "Why do you usually listen to this track?", was based on the 16 statements used

by Greasley and Lamont [1]. Unlike in the original paper where they were asked to choose all that apply, each participant was asked to rank the reasons from 1–5 (1 = most relevant, 5 = least relevant). Due to a technical glitch, only 70 participants (20% of the participants) filled out this question. As we were not aware of an existing set of questions that had been used to specifically understand why people download music and save it rather than listen to it, we created our own question, "Why did you download this track?", and provided 15 options covering three areas: (1) they liked it or something about it, (2) they heard it somewhere or (3) they were recommended it by someone else or it had meaning for them. Participants could tick all that applied of these options. Finally, we asked when they had last heard the tracks.

The third section included questions about their music listening habits and demographic characteristics. They only answered these questions once. To measure musical engagement we used the Goldsmiths Musical Sophistication Index, Musical Engagement Subscale [28]. This nine-item scale ($\alpha = .872$) measures the amount of musical engagement or the amount of time and money people spend on music or music-related activity.

The questionnaire closed with two items that checked how participants had completed the survey. We asked whether they had skipped any tracks before they settled on the track to answer about and whether they had listened to the tracks while completing the questionnaire. Finally, participants could provide comments or feedback about the survey.

The survey, that can be found in full in the supporting information (S1 File), was created and administered online using Qualtrics [29].

## Procedure

People with and without musical training were invited to participate via social media sites and over the internet. A snowball sampling [30,31] recruitment procedure was used with direct email invitations sent to target participants from the authors' professional and social networks. This allowed the researchers to specifically target different populations, i.e. international and older participants, to help reach a wider audience. Participants were also invited to forward the email invitation to others they thought might be interested. In addition, the link to the survey was shared on social network sites (Facebook and Twitter). Using social media as an adjunct recruitment strategy has been found to be a successful and cost-effective recruitment style [32].

When participants opened the survey link they were given information about the study and an online consenting process was activated. Once participants had given consent they could begin the online questionnaire.

## Analysis

We analysed our quantitative data using simple distributions, frequencies and means using SPSS (IBM SPSS Statistics for Macintosh, Version 25.0) [33] and R (Microsoft R Open 3.5.1) [34]. All visualizations were created using Tableau Desktop and are hosted online at Tableau Public. All of the raw data is freely available and can be downloaded from https://miguems.github.io/dotheshuffle/, a webpage specifically designed to accompany this paper.

Following Braun and Clarke [35], thematic analysis was used to identify categories and themes that represented the data. Overall, categories were identified which became the first codes (level 3, L3). These codes were then grouped into themes (level 2, L2). Finally, themes were merged to develop higher-level themes (level 1, L1).

After reviewing a subset of the data (72 responses in total), two different coders (Coder 1 and Coder 2) independently identified categories to represent key elements of the data by creating level 3 (L3) codes. Both coders then compared their codes and developed larger

categories (L2) to be incorporated into a codebook. (See the supplementary information (S2 File) for the full codebook with definitions and examples). A new coder (Coder 3) used the codebook to code the entire data set. To verify that the final coder had interpreted the codes as intended, the codes assigned by Coder 3 were compared to the original codes assigned by Coder 2. Coder 2 assigned more codes (L3 and L2) overall ($n$ = 436) than Coder 3 ($n$ = 364). 66% of Coder 3's codes were in agreement with Coder 2's codes. This calculation included the times when Coder 3 did not assign a code. Following this, the research team discussed any discrepancies between the codes and the final coding of all the data was then finalized. Following this, higher level themes (L1) were developed.

## Data visualization

Several visualisations were created to represent the open-ended responses collected. The visualizations are available online on the project website [https://miguems.github.io/dotheshuffle/]. Developed with Tableau, they include several data filters to enable a more detailed view into distributions of the identified codes by sex and use of music within their professional lives. While this capability is not extensively used in this paper, the interactivity provided by the filters allows the exploration of relationships of the different codes with certain demographic factors and responses to the closed questions.

## Survey experience

The median duration of survey completion was 17 minutes. 278 out of 322 participants (86%) stated they had not skipped any tracks (after clicking shuffle) before writing about the tracks. 12% skipped one track, either after the first or second time they shuffled their device, while only 2% skipped for both tracks. The average enjoyment level was 4.34 ($SD$ = 0.79) out of 5 for the tracks discussed by participants that did not skip a track, while the average for the tracks discussed by those that skipped one or more tracks was 4.33 ($SD$ = 0.66). Further investigation showed that there were no relevant differences in the distribution of assigned codes, as illustrated by the skipped track filter in the interactive visualisation. Therefore, responses by participants who had skipped tracks before responding were combined with those by participants who had not skipped tracks in the analysis. The majority ($n$ = 387) of the 397 participants answered all the open-ended questions for two different tracks. The responses from the participants that only answered the questions for one track ($n$ = 10) were also included in the analysis. Even though participants were not explicitly asked to do so, 83.4% (out of 322 participants who answered this question) reported listening to at least one of the tracks when writing about it. 66% (out of 322 participants who answered this question) had heard the track that they wrote about during the last month, suggesting that for the majority of the participants the tracks were relatively fresh in their minds.

For the question, "Do you have a relationship with this track? Whether it's 'Yes' or 'No' please explain", participants responded that they did not have a relationship with a large proportion (about 50%) of the 784 total tracks discussed. Interestingly, some of those participants who answered "no" describe aspects that fit with our understanding of the term 'relationship' within their responses (e.g. enjoyment, used for a specific purpose, associations etc.). With some of the participants responding "no", our findings suggest that our understanding of the word 'relationship' in the context of music listening may not be representative of how a substantial group of our respondents think about the music they listen to, associating this phrase with an emotional response rather than an associational one.

## Results

### RQ1: Why do people download and listen to music? Because they like it

The top four reasons for downloading the tracks were that the participants had liked the artist ($n = 340$ out of a total of 1887 options chosen, 18%), the way it sounded ($n = 329$, 17%), the album ($n = 232$, 12%) and the genre ($n = 221$, 12%). Choices according to basic labelling information–liking the artist, album and genre–accounted for almost half the reasons for why participants downloaded the tracks (47%). All top answers fit within our broad category of choosing to download a track because the participant liked it or something about it (Fig 1).

The top five ranked reasons for listening to the track were "Because I really like listening to it" ($n = 28$, 20%), "To help carry out/enhance the activity I was doing ($n = 19$, 13.57%), "To accentuate an emotion/mood" ($n = 13$, 9,29%), "None of the above" ($n = 13$, 9.29%) and "To listen to the lyrics" ($n = 11$, 7.86%). While this data was only collected from 70 participants due to a glitch in the online survey, these top five most relevant options make up 60% of the total options ranked by those 70 participants (Fig 2). We performed a Friedman's test to determine if the differences between rank positions were statistically significant, finding that there were significant differences in how these options were ranked $\chi2(12) = 167.96$, $p < 0.05$. Post hoc analysis using Wilcoxon signed-rank tests was conducted, with a Bonferroni correction applied ($p < 0.01$), to test for significant differences in the top five ranked options. "Because I like listening to it" was ranked significantly higher than "To help carry out an activity" ($Z = -3.566$, $p < 0.01$). However, "To help carry out an activity" and "To accentuate a mood" were not ranked significantly differently ($Z = -1.254$, $p = 0.21$). "To accentuate a mood" was ranked significantly higher than "None of the above" ($Z = -5.337$, $p < 0.01$) that was in turn ranked significantly higher than "Listening to lyrics" ($Z = -3.65$, $p < 0.01$).

784 tracks were included in the overall analysis. The majority of the tracks (91%, $n = 719$) were rated as enjoyable (enjoy ($n = 336$, 43%) and really enjoy ($n = 383$, 49%)). Few tracks (9%, $n = 67$) were rated by the participant as not enjoyable (1% really did not enjoy ($n = 7$), 2%

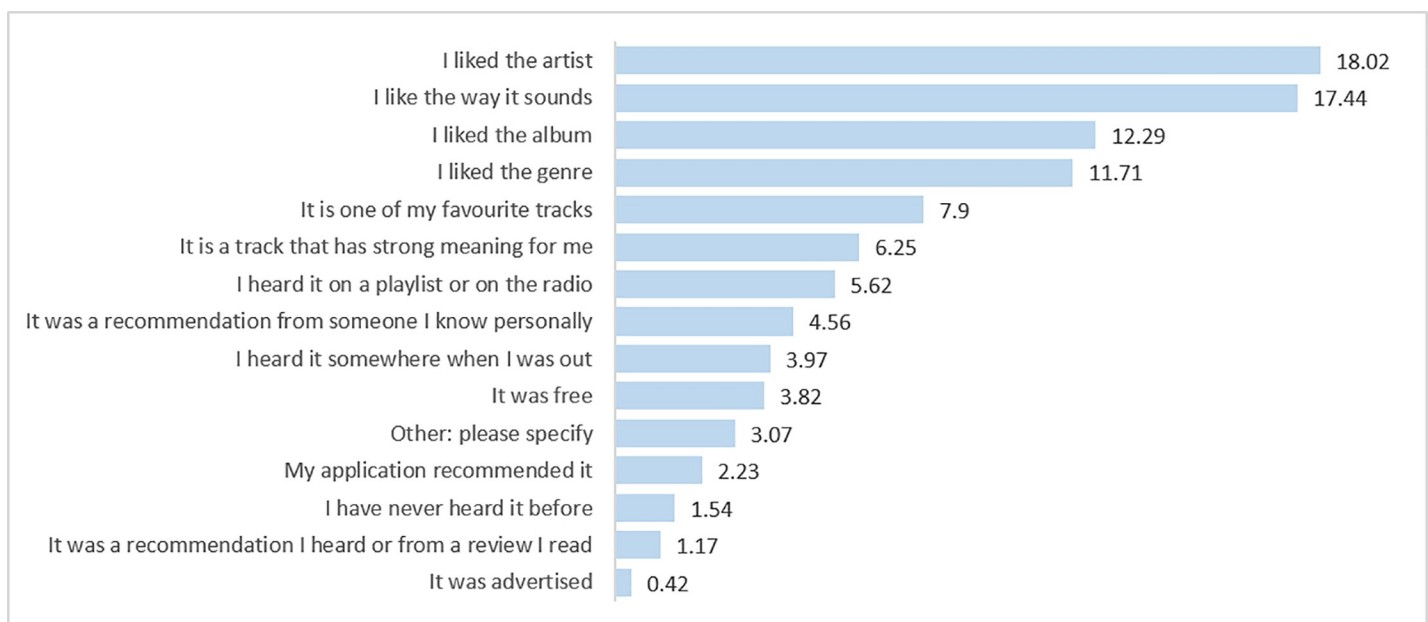

**Fig 1. The most common reasons for downloading the tracks ordered by %.**

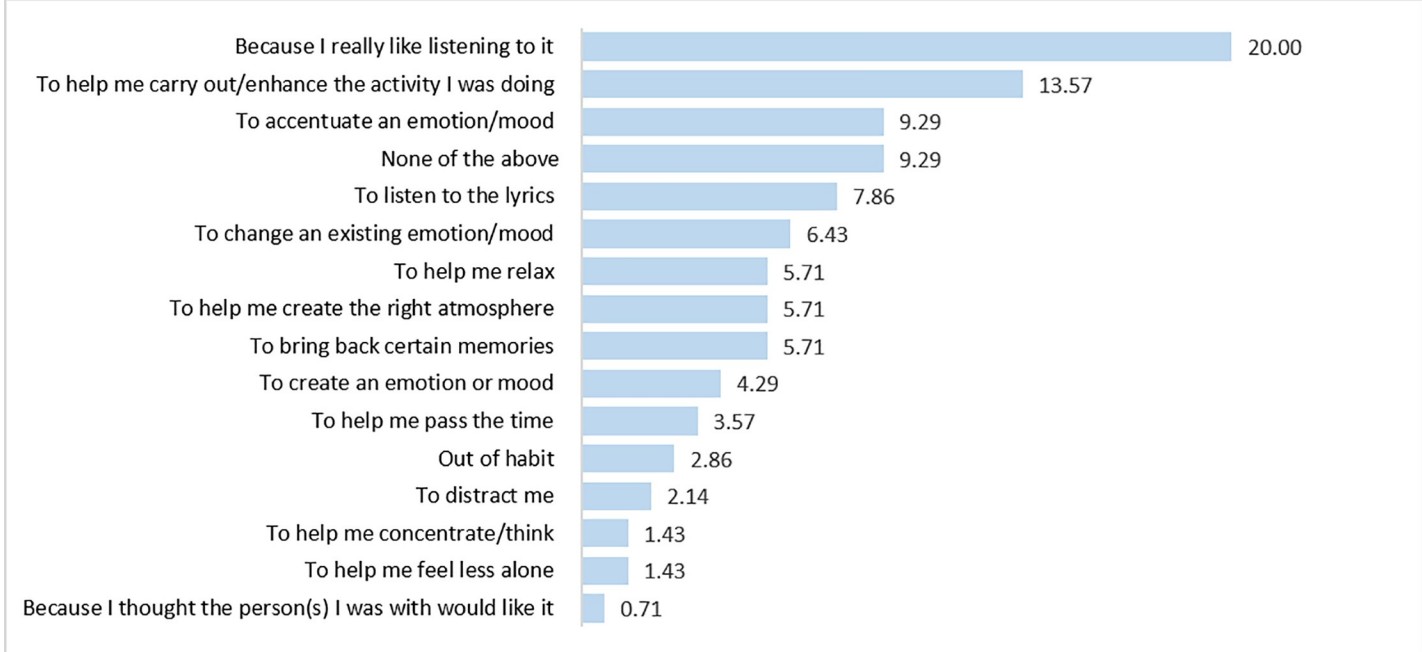

**Fig 2. The first ranked reasons (ordered by % of total) chosen by the 70 participants who answered why they usually listen to the track.**

did not enjoy (*n* = 14) and 6% neither enjoyed/not enjoy (*n* = 44)). Ten tracks were not rated for enjoyment.

### RQ2: Which aspects of the music listening experience are prioritized when people talk about a track on their device? The informational context around it

Each participant shuffled their device twice and gave the title and artist of the tracks that came up. A total of 784 tracks were identified by the 397 participants who completed this portion of the questionnaire. Responses to all of these 784 tracks were included in the analysis. Of these 784 tracks some were duplicates. For example, three participants discussed *Castle on the Hill* by Ed Sheeran. (The track titles, artists and extracted Spotify data can be found in the supplementary material (S3 File)).

Each participant (*n* = 397) answered four open questions about two separate tracks: "What's the first thing that comes into your mind about this track?", "Why did you choose to save this track to your device?", "Why do you enjoy or not enjoy listening to the track?" (which followed the enjoyment rating) and "Do you have a relationship with this track?". An initial reading of the responses suggested that participants often repeated information when answering the different questions and also referred back to previous answers when talking about the same track. The responses to the four questions were coded as one unit in order to avoid redundant coding within one participant's responses about a track. The majority (97%) of participants answered these questions for two different tracks. Those that only answered the questions for one track were still included in the analysis. Each participant that answered the questions for two tracks (*n* = 387) had one paragraph of text for each track and each paragraph was coded separately. Differences between people who use music in their professional life and those that do not were investigated. There were no differences found in people responses to the closed questions and the themes identified from the open-ended questions.

As mentioned above, the codebook consisted of 3 levels of codes. The highest level (L1) had 4 higher-order themes. The middle level (L2) had 14 categories while the lowest level (L3) had 74 codes with a minimum of four and a maximum of 505 occurrences. Occurrence refers to the number of times that a code was discussed in relation to any track. For example, if the code "artist" was identified in one participant's response to their first and second track this would count as two occurrences coded as artist. In total there were 4,182 code occurrences. See Fig 3 for a breakdown of the code hierarchy.

Four main higher-order (L1) themes were developed: Associations, Characteristics, Evaluations and Responses Induced (the total occurrences for each L1 theme are given in Fig 4). These themes address which aspects of the listening experience people prioritise when talking about a track on their device. Associations, as the most common type of response (with 1,946 occurrences out of the total 4,182 occurrences), were prioritised over listeners' characterisations and evaluations of the track and how they responded to it. The themes developed from the analysis showed that participants based their evaluations, characterisations and responses not just on the music itself but also on the music-informational context around it; the artist, album, and genre.

## Associations

The Associations higher order theme was used when the respondent related the track to something or someone outside of the musical track itself (e.g. "I did think of its brilliantly directed and acted music video quite a lot after watching it as it is quite a powerful video"). We identified six different L2 categories within Associations: Track Identifiers, Memory, Activity, Person, Other Media Forms, and Imagination (Fig 5 shows the breakdown of all the L2 themes). Responses were coded as Track Identifiers (the second most common code with 625 occurrences making up 15.0% of the overall codes and a third of the Associations codes (32.1%)) when the respondent associated the track with the album, artists, era, genre, or version of the track (e.g. "I have all of The Beatles' albums saved because they're obviously amazing."; "It's a fairly early Prince track from the era when he played everything himself"; "This is the O.S.T. version."). A response was coded as Memory (531 occurrences) when the respondent associated the track with the memory of discovering the track, an event, the last time they heard it, a place, a time or stage in their life (e.g. "Remembering the last time I heard it at a party and what happened that night"; "Now every time I listen to it, it reminds me of dancing like crazy at the concert"; "If I stop to think about it I can associate events with it (playing it on the piano at home, listening to it in a concert once half a life-time ago, and imagining the performer groaning his way through some of the other tracks on the disc)").

Participants associated their tracks with an Activity (301 occurrences; e.g. "I enjoy calm solo piano music to either read or sleep to"; "Its catchy to walk/travel to") or a Person (206 occurrences; e.g. "Reminds me of my boyfriend when he's away from me"; "The Ozarks were a favourite group of a cousin, who I loved"). Participants also associated the tracks with Other Forms of Media (176 occurrences) for example the music video, movie, television show or musical the track appeared in (e.g. "I have been to see the musical which it is from (Phantom of the Opera) and so bought the soundtrack"; "I associate this song with the TV show called Glee."), the playlist the track was from or the application it was found on (e.g. It might be another Discover Weekly, and I sometimes mooch around the suggestions on Spotify, or 'similar artists' links.").

Finally, another L2 category identified was Imagination (107 occurrences). A response was coded as Imagination when the respondent described the track through a descriptive narrative which is not a retelling of a memory (e.g. "The intro is very haunting so I picture a dark club

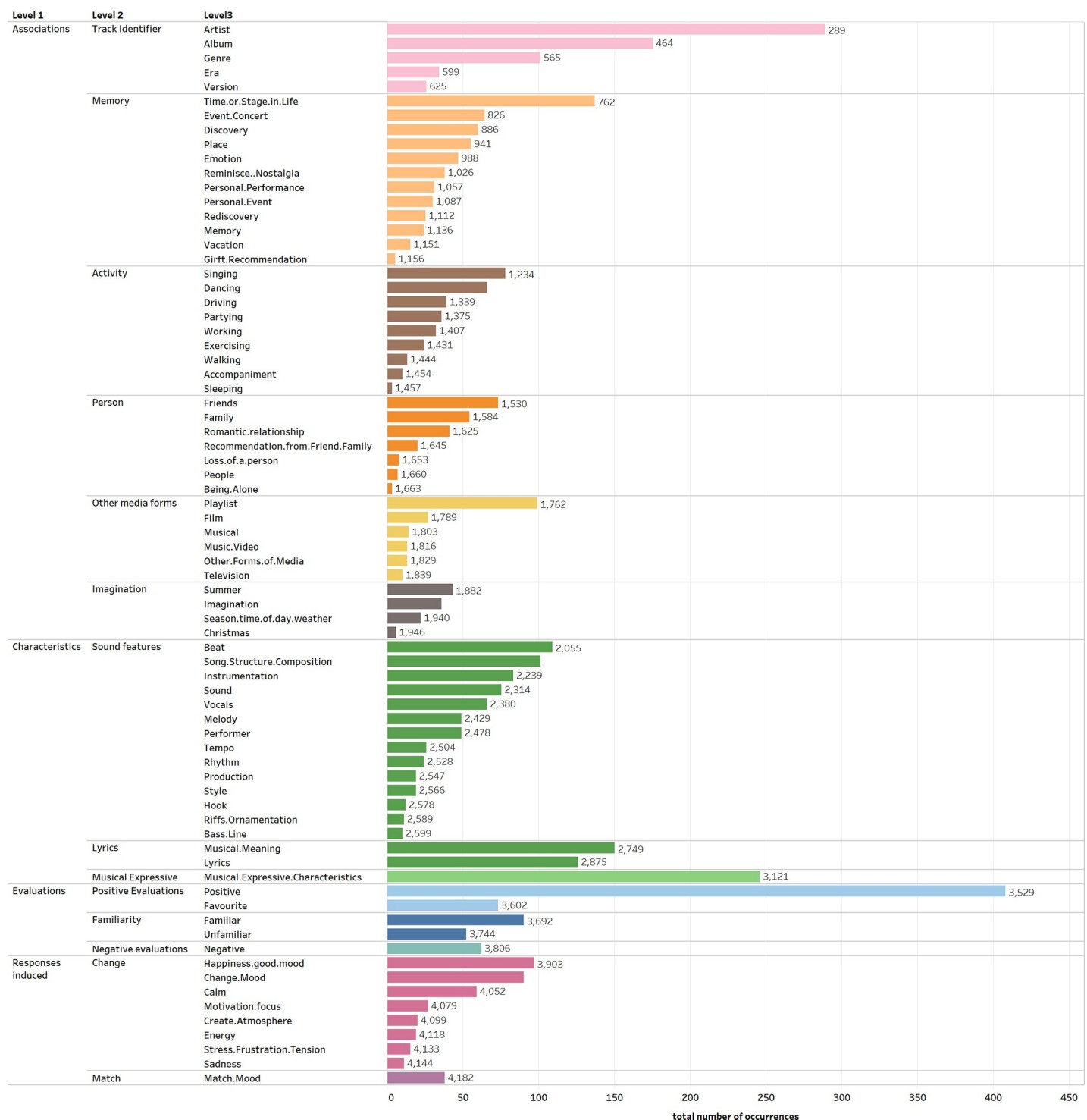

**Fig 3. All codes by each level.** The number of occurrences that were coded for each level 3 code.

with red lights that flicker through the club"; "I feel I am at a classic bar or lounge where I am smoking a Cuban cigar and sipping a drink with my friends. At the same time a group of female dancers dance according to the beat of this track"), or a description of the weather or

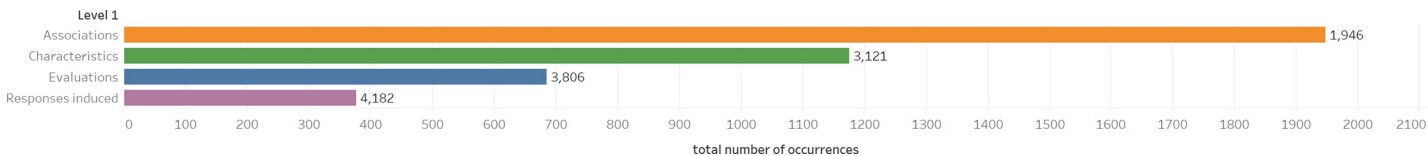

**Fig 4. The number of occurrences that were coded for each level 1 code.**

season associated with the track (e.g. "It's quite summery and makes me think of hot summer nights"; "The first thing that comes to mind is a warm summer day and feeling the heat on my face."; "Spinning in a field- arms out enjoying the sun").

## Characteristics

Characteristics had 1,175 occurrences (28% of the total codes) and was used when the respondent described or talked about the track by characterising it by how it sounds using expressive adjectives (e.g. lively, lovely, peaceful) or by describing the more technical aspects of the track (e.g. the beat or the lyrics). These were divided into three sub-themes: Sound Features, Lyrics, and Musical Expressive (Fig 5).

The category with the most occurrences overall and in this higher order theme was Sound/Musical features with 653 occurrences. This category makes up 15.6% of the overall codes and half (55.6%) of the codes under the Characteristics higher-order theme (Fig 3). Responses were coded as Sound/Musical Features when the respondent described the track using technical terms like beat, melody or production (e.g. "It's got a wonderfully syncopated bass line. . ."). With 14 different technical musical features identified, there is a wide range in the variety of the technical terms participants referred to. Some specific features identified were the production (e.g. "Troye soulfully sings the song's simple yet effective lyrics complimented by some excellent production which is paced very well feeling subtle at times and then more dramatic as the song goes on"), hook (e.g. "Melody serves as a very effective hook") and Song Structure/Composition (e.g. "It's one of those great tracks that builds to a climax and doesn't disappoint") of the tracks.

Lyrics (276 occurrences) were coded separately from Sound/Musical features (e.g. beat and melody). This was because a large majority of participants talked about the lyrics specifically and did so separately to other sound features. A response was coded under Lyrics when the respondent talked about the track by mentioning or describing the lyrics or the lyrical meaning (e.g. "It has a cool meaning behind the lyrics"; "I think the lyrics are really meaningful and

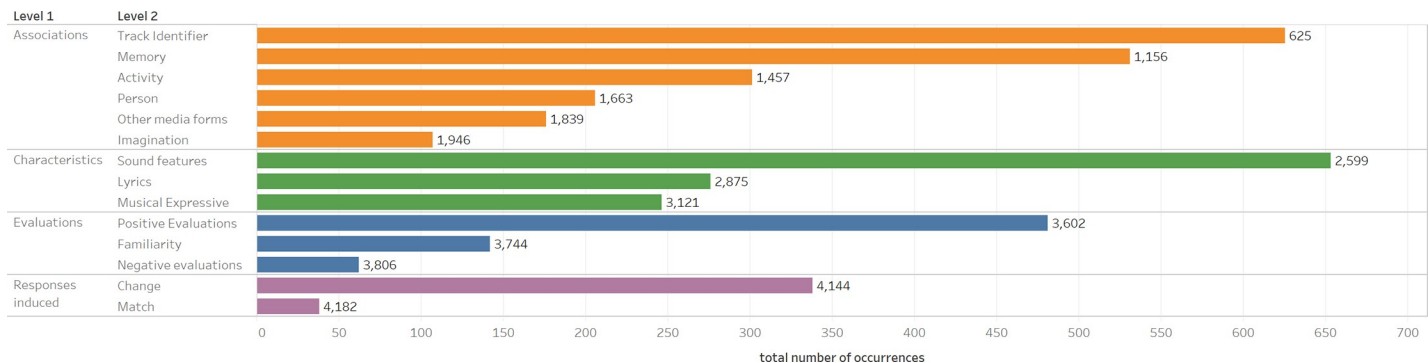

**Fig 5. The number of occurrences that were coded for each level 2 code.**

poetic."). Also under the Characteristics theme was the category Musical Expressive (246 occurrences). Responses were coded as Musical Expressive when the respondent used a descriptive adjective to describe the track. The adjectives used varied widely (e.g., "Bouncy", "Melancholy", "Vivacious", "Ploddy Dirgey").

The variety in the identified musical features and the musical expressive words speaks to the variety in the music represented in this sample as well as the high level of general musical expertise and the richness of music description of music listeners today. People seem to use a variety of everyday adjectives and technical terms to talk about their music. This diversity shows the vastness in the music as well as people's characterisations of and interactions with that music.

## Evaluations

There were 685 occurrences of Evaluations. A response was coded within Evaluations when the respondent evaluated the track, an aspect of the track, their relationship to, or familiarity about the track (e.g. "No—don't like arctic monkeys"). These were divided into Positive Evaluations (481 occurrences), Familiarity (142 occurrences), and Negative Evaluations (62 occurrences) (Fig 5).

A response was coded as Positive Evaluation when the respondent made a positive evaluation of the track or an aspect of the track, for example what the song is associated with, the artist or TV show it is from (e.g. "I love musicals and Rent is spectacular, not just for the phenomenal tracks for the meaning and characters, the words, and a great sing-a-long"; "I love Beyoncé, her music makes me feel like anything is possible"; "I like the track and I really like the singers voice") or a musical feature (e.g. "Peaceful gorgeous song, rich voice, really simple composition"; "Sassy good sassy song for the club). However, not all evaluations of people's music on their device were positive with 9.1% (62 occurrences) being negative evaluations (e.g. "I want to skip it. When I got the track, I loved the track and now I am over it. I am just kinda over it, it is a bit annoying now").

Participants reported different reasons for negative evaluations of tracks. One was changing taste, as seen in the previous example. Another was the fit between the music and their own state (e.g. "Sometimes I do enjoy listening, sometimes I don't—depends on my mood and the situation, as rock 'n' roll is not always what I like"). Some negative evaluations resulted from comparisons to other tracks by the same artist or within a certain album (e.g. "I like it but not as much as some of her other tracks") and to other aspects within the same track (e.g. "Terrible chorus. Verses are decent, so is the beat during the verses but the chorus is just so boring"). Participants also gave negative evaluations of how others might judge them for having the track (e.g. "so I should be a bit ashamed of liking them. I'm not ashamed. Not much.") and also gave clear neutral or negative judgements of the track (e.g. "This track is annoying"; "Boring. Doesn't really appeal to me").

## Responses induced

The final higher level theme, Responses Induced, was coded the least frequently with 376 codes (8.9% of the total codes) and was used when the respondent described or talked about their response to listening to track, how it made them feel or what it made them do (e.g. "Also, I feel calm when playing this track").

Under the Responses Induced higher-order theme two different categories were developed: 90.0% of Response Induced codes (338 occurrences) were coded as Changing Response, while 10.1% (38 occurrences) were coded as Matching. A response was coded as Changing Response when the respondent described or talked about how the track changes how they feel or the

atmosphere (e.g. "This track boosts up my energy levels and gives me the mood to work and chill at the same time"; "Makes me feel relaxed. Evokes a calm emotional response."; "This song makes me stand a bit on edge"). A response was coded as Matching when the respondent talked about the track matching their expectations, as useful for listening to when they wanted to match a mood, or stating that it did or did not match their current mood (e.g. "It's more upbeat than the last one, and less dirgey, more uplifting, particularly the rising harmonic progression in the middle 8, so more suited the mood I'm in which is positive right now."; "It's a good song I just am never in the mood for it"). This theme may be the most represented in previous literature but our analysis and method suggests that people do not prioritise their emotional responses to their music over their evaluations, characterizations, or associations with that music.

## Results summary

Some themes in our analysis have been discussed in previous research. For example, associations with specific memories [19], activities tracks are used for [26], or expressing the different types of emotional responses tracks can induce [11]. However, other themes showed a richness previously underrepresented in the literature. For example, the many ways people associate tracks to information outside of the music itself and the variety in the way people characterise music. Moreover, the context considered when discussing the track was not just that of the music-informational context (e.g. the album or artist) but also the wider media the track may have been associated with (e.g. the music video, musical or film the track appeared in). People also associated a track with a setting and used their imaginations to create a scenario that the music represents. Finally, by using the *Shuffled Play* method, we were able to elicit a variety in people's responses. This included some tracks participants did not enjoy or did not like specific aspects of. This helps highlight the nuanced way in which people talk about music.

## Discussion

Over 70% of people in our study choose to download music based on what they like about it: who the artist is, the way it sounds and/or which album it is from. However, because some people also download music onto their device for other reasons–a specific event, enjoyment of an artist's previous tracks, or using it for an activity–they do not necessarily like all their downloaded music. This highlights the importance of our method. Using *Shuffled Play* enables an exploration of these other types of relationships with music. Compared to previous experimental methods (i.e., researchers selecting the music themselves or asking about favourite pieces), this method is more naturalistic and similar to how people listen to music on their devices. This method also allows participants to cover a wider range of music that might be meaningful to the participant.

When considering why participants listen to a given track, comparing our results to the same question asked in Greasley and Lamont [1] we found the same top reason selected: "Because I really like listening to it". However, none of the other top five overlapped. Perhaps the reasons lie on the different methods used: we asked participants to shuffle their music libraries whilst Greasley and Lamont [1] used experience sampling to ask people about music they heard at different moments during the day. In fact, one of the top five ranked reasons in this study was "None of the above". By analysing responses from both open and closed questions, a better understanding of the potential reasons this option was chosen so often was achieved.

The most coded L3 code was Positive Evaluation (481 occurrences) and the top ranked reason for listening to the track was because participants liked it. Even though this may seem

obvious, it is important to highlight. From this study however, it is difficult to ascertain whether "Because I like it" is an independent category or an umbrella term used that incorporates the various other more fine-grained motivations usually associated with music listening like mood regulation. It may also be through enjoyment that music can have other functions. For example, one can speculate that if people do not enjoy the music in the moment of listening, then it might not help them use music in the ways we often hear about (relax at that time, run faster, etc.). Future research could further explore the potentially moderating effect of enjoyment.

The results of this new method of data collection also highlight the importance of music-informational context as a factor when people describe their relationship with the music they have saved on their devices. Recent research has begun to highlight the relevance of music-informational context on people's musical choices and judgements. For example, the linguistic fluency of the song title and artist [36] and the information included in program notes [37] have been found to affect people's aesthetic judgements of music at least in some cases.

In our study we see that one important music-informational factor is the artist. This contextual information itself may override certain sound preferences: artist is the second most frequent L3 code (and is under the Track Identifier L2 theme). The results highlight the relationship that listeners have with particular artists as a driving force when describing and making judgments about music. It has previously been shown that the perceived prestige of the artists of a track can change the judgements people make about the same musical pieces when they are told they are different [21]. A performer's status can bias the evaluation of musical stimuli [38] and empathy with an artist was found to be an important factor in aesthetic experiences of music and poetry [39]. The importance of specific people can also be seen in classical music as canonical composers have dominated music history, concert platforms and societal preferences for centuries. Our results provide a similar and supplementary result, that people prioritise the artist not just when judging but also when talking about music. By using a completely different method–primarily descriptive compared to the neurological and quantitative methods of the previous literature mentioned–we come to a similar conclusion: people's feelings towards the artist and the informational context around the track can drive their discussion of the music, their preferences for it and whether or not to download and listen to it.

People seem to be aware of the importance of the music-informational context in their music listening and downloading experience (cf. [17]) and in our open-ended question data we find that people can be critical and acknowledge the importance of the informational context by discussing their liking of the artist or album as their reason for choosing the music. They can like the artist but not the track or they can like the track but not the artist. Very few research studies acknowledge the complex relationship between preference for an artist and preference for a particular piece of music in terms of everyday music listening experiences. It may also be the case that people have a two-step process when thinking about why they download music that might be indicative of how people search and find music online. First decisions might be made based on genre or artist preference and following this judgment, preference around the sound and the music itself may be made. This study, along with the few previous studies mentioned, exposes the need for more research investigating this important and complex relationship.

Previous research has also focused on the emotions music can evoke, arguing that this may be an important function of everyday music listening [10,11]. This study shows that this is not an aspect of the music listening experience people consistently prioritize in a *Shuffled Play* situation. People describe the track, characterizing what they like and do not like about it and what it reminds them of before they describe how it makes them feel. They also seem to have a wide variety of terms that they use describe the music, whether it be musical features, descriptive adjectives, or an image or scenario. Juslin's [40] model proposes a theory that visual

imagery acts as a potential mechanism for why music evokes emotion. The identification of the Imagination and Musical Expressive themes in our data set might speak to the importance of visual imagery and imagination in people's experiences with music. People also take into consideration the lyrical content of the piece as an important element to prioritise when describing a particular piece of music. All of these findings show the range of characterizations of the music that people use when discussing music in everyday life.

While some of our results are in line with previous literature, this study uses a novel method that refocuses or reframes previous findings and thinking. This study has highlighted that enjoyment is important and should not be ignored but rather emphasised when discussing people's relationships to music. The functions and emotions felt when listening to music can be highly affected by, and dependent on, the participants' enjoyment as well as the informational context around the music. It is not only situational context, which has been discussed in previous literature, but also the information around the music itself that seems to be most important.

Our study offers resources to continue or extend explorations of shuffled music listening. In particular, we have made openly available the datasets with the responses, codes and demographic information, and we have created a set of interactive visualisations that can be freely accessed online and experimented with. For example, research questions about the relationship between different music genres and characteristics and the way they are described are beyond the scope of this paper but the data set lends itself to such analysis. Similarly, the visualizations available could be used to further explore the relationships between the identified themes and particular demographic factors. With our unique method, along with our interactive visualizations, we were able to explore in a novel way why people download and listen to music and which aspects of the music listening experience are prioritized when people talk about a track on their device.

## Limitations

The shuffle function is used with downloaded music by some music listeners, but we do not assume that this way of music listening is representative of music listening habits across the board. We also consider that the results of this study are a reflection of the specific *Shuffled Play* method utilized in this study. Moreover, the shuffle function, while useful in one respect, means that only a small portion of people's music collections were investigated. Furthermore, the shuffle function is only semi-random—the tracks that are displayed were not selected in a truly random way. Setting aside the process of the shuffle algorithm embedded in Spotify and other music listening applications, 44 participants, out of the 322 that answered this question, skipped the track which was chosen through the shuffle method for either one or both of the tracks (of these only five skipped the app-chosen track for both responses).

In addition, some applications do not have a shuffle function for a whole music library (e.g. Spotify). In this case our method limited the tracks selected to no longer represent participants' full library but rather a selection of their library or playlist. In addition, the participants themselves, which primarily represents female, young adults, and people from or residing in the UK, cannot be assumed to generalise to all contexts and cultures. Nevertheless, despite these limitations and helped by our large sample size, the data set includes a wide variety of tracks and participant descriptions.

Due to a glitch in the online survey, only 70 participants ranked the reasons why they listen to 140 tracks. The original paper by Greasley and Lamont [1], from which this question was based on, included 25 participants. Therefore, the interpretation of these current results, though taken from a smaller subgroup of our sample, is significant in comparison with other studies in this field.

A potential area of bias in our design was in how the questions were structured in the survey. We asked questions about why they had the piece on their device that may have prompted a more practical way of discussing that music. To mitigate this, these questions were asked after all the open-ended questions had been answered. However, the task itself may have prompted more practical rather than emotional responses as we did not specifically ask them about a song that they had an emotional connection with.

Previous literature has also discussed the importance of context when discussing everyday experiences with music [1,3,6]. People chose music to put on their device for use in different contexts, which is seen in some of the responses from our participants who associate a track with a certain activity. However, the shuffled listening situation used in this study disregards this, as the track chosen is semi-random.

## Implications for future research

The understanding that the music-informational context plays a significant role in people's music listening experience should be considered when addressing other questions about musical preference, functions of music listening and musical judgements. Furthermore, research in music therapy and music in health suggests that giving people an opportunity to experience music in and of itself without a specific functional outcome is important and valuable. However, there is little research explicitly investigating where the aesthetic experience and enjoyment of music sits in relation to people's everyday experience of music and its use in health contexts.

While this study included responses to some music that participants disliked, the participants liked the majority of tracks they discussed and no data was collected on the tracks that were skipped. A modification of this method could usefully intervene at the moment of skipping tracks to understand why people might dislike a track or in what contexts and why people might skip a track when listening to music using *Shuffled Play*.

Future research could also explore the musical features and their relationship to people's descriptions of the music. We have included with this paper the extracted Spotify data (such as the track title and artist, the average bpm, and loudness) in the supporting information (S3 File). This data was extracted using Spotify's "Sort Your Music" application (http://sortyourmusic.playlistmachinery.com/). Data like this could be used to help explore this relationship, as has been done for preferred music by Knox and MacDonald [41].

## Conclusion

The novel method used in this study helps to illuminate the variety of ways people discuss and talk about music they have curated or collected on their devices. While the specific findings are not revolutionary, the new method and the mix of response type data used allows for the illumination of the diversity and subtlety of personal listening experiences. Overall, people listen to and download music because they like it. This straightforward but important observation is usually glossed over in the understanding of music listening habits, particularly when looking at the impact of functional music listening. Moreover, what people like about the music is found here to depend on a variety of important factors, mostly those outside of the musical content itself. People make critical judgements about many different aspects of the music and seem to prioritize the music-informational context along with the characterization of it over their emotional responses to it. This gives insight into what drives participants' music preference and enjoyment. These findings may contribute to improved understanding of how music may be more appropriately used, and function better, in health, educational and social settings.

## Supporting information

**S1 File. Full questionnaire.** Administered online through Qualtrics.
(PDF)

**S2 File. Code book.** Finalized code book used by both coders.
(PDF)

**S3 File. Track and extracted Spotify data.** Track information given by the participant and track data extracted from "Sort Your Music" application.
(PDF)

## Acknowledgments

We would like to thank Nordoff Robbins and the Royal College of Music for supporting the project. We would also like to thank Victoria McCrea for her help with survey design, participant recruitment and data analysis; Megan Kibler and Chloe Parks for their help with data analysis; and Li-Ching Wang and Owen Coggins for help and advice throughout.

## Author Contributions

**Conceptualization:** Katie Rose M. Sanfilippo, Neta Spiro.

**Data curation:** Katie Rose M. Sanfilippo, Neta Spiro, Alexandra Lamont.

**Formal analysis:** Katie Rose M. Sanfilippo, Neta Spiro.

**Funding acquisition:** Neta Spiro.

**Methodology:** Katie Rose M. Sanfilippo, Neta Spiro, Alexandra Lamont.

**Project administration:** Katie Rose M. Sanfilippo.

**Supervision:** Neta Spiro, Alexandra Lamont.

**Visualization:** Miguel Molina-Solana.

**Writing – original draft:** Katie Rose M. Sanfilippo, Neta Spiro, Alexandra Lamont.

**Writing – review & editing:** Katie Rose M. Sanfilippo, Neta Spiro, Miguel Molina-Solana, Alexandra Lamont.

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
