## [Decision Letter · Decision Letter 0]

14 Oct 2019

PONE-D-19-17641

Do the Shuffle: Using shuffled play to explore reasons for music listening

PLOS ONE

Dear Neta Spiro,

Thank you for submitting your manuscript to PLOS ONE. After careful consideration, we feel that it has merit but does not fully meet PLOS ONE’s publication criteria as it currently stands. Therefore, we invite you to submit a revised version of the manuscript that addresses the points raised during the review process.

In particular, the mixed methods design of your study needs to be specified and integration needs to be addressed.  It is recommended that you integrate your quantitative and qualitative findings narratively and via a joint display.  Furthermore, the novel contribution of your work should be elucidated.

We would appreciate receiving your revised manuscript by 11 January 2020. To enhance the reproducibility of your results, we recommend that if applicable you deposit your laboratory protocols in protocols.io, where a protocol can be assigned its own identifier (DOI) such that it can be cited independently in the future. For instructions see: http://journals.plos.org/plosone/s/submission-guidelines#loc-laboratory-protocols

We look forward to receiving your revised manuscript.

Kind regards,

Sarah E.P. Munce, PhD

Academic Editor

PLOS ONE

**Journal Requirements:**

**Additional Editor Comments (if provided):**

Thank you for your submission to PLOS ONE. Please review and address all of the comments of the individual reviewers. In particular, the authors need to specify the mixed methods design of this study and integrate the findings of the qualitative and quantitative components (narratively and via joint display). The authors should also clearly elucidate the novel contribution of this work.

**Comments to the Author**

1. Is the manuscript technically sound, and do the data support the conclusions?

Reviewer #1: Yes

Reviewer #2: Yes

Reviewer #3: No

Reviewer #4: Partly

2. Has the statistical analysis been performed appropriately and rigorously? 

Reviewer #1: Yes

Reviewer #2: Yes

Reviewer #3: Yes

Reviewer #4: Yes

3. Have the authors made all data underlying the findings in their manuscript fully available?

Reviewer #1: Yes

Reviewer #2: Yes

Reviewer #3: Yes

Reviewer #4: Yes

4. Is the manuscript presented in an intelligible fashion and written in standard English?

Reviewer #1: Yes

Reviewer #2: Yes

Reviewer #3: Yes

Reviewer #4: Yes

5. Review Comments to the Author

Reviewer #1: In this paper the authors examined factors that lead individuals to download and/or listen to specific music selections. Rather than responding to pre-selected music examples or selecting their own music examples, participants responded to two pieces that were generated by their device’s “shuffle” feature. This was an innovative design twist that allowed for a potentially richer set of data, tapping into choices that at once represented the participants’ own volition yet did not necessarily represent their strongest preferences.

Overall, I found this to be an interesting paper that nicely balanced precision necessary for survey inquiry with the more natural context-dependent reality of personal music choices. As suggested by the authors, the findings were not ground-breaking in their novelty; rather, they underscored the fluidity and diversity of the personal music listening experience.

I do have several comments and suggestions for the authors. These are largely focused on aspects needing greater clarity or consistency, however, as such, are consequential for helping the reader to understand and interpret the data and interpretation presented.

1. Abstract - The title/abstract makes reference to both exploring “reasons for music listening” and the innovative aspect of the shuffle play method. Each of these issues receives different emphases throughout the paper making it somewhat unclear exactly which is the main focus of the paper. Was the main takeaway the insight into listening or the viability of the method?

2. Abstract – To what degree might “because they like it” subsume the various other more fine-grained motivations usually associated with music listening (e.g., mood regulation)? Is “liking” really a different category of reason or a broader umbrella term used by the participants to describe a range of listening experiences?

3. P. 3, line 46 – “of the time” seems a bit vague. Does this refer to any time? Some specific subset of time?

4. P. 5, line 112 – This is unclear. Was shuffle or travelling the largest category? Category of what?

5. P. 6, line 121 – Here and throughout, I am not sure this represents a “mixed methods” approach as formally defined by sources such as Creswell (2010). The use of both closed- and open-ended questions provided two different types of data, however both were the result of a large-scale survey method (and, in fact, both gathered from the same group of people, using the same instrument, at the same time). Consider using a different term to convey use of this mixed response type.

6. Table 1 – This breakdown of the sample is confusing as the different subsamples have not yet been explained in the report. The note is only marginally helpful because it references a survey item not yet presented in the method.

7. Table 1 - The diverse array of countries represented is impressive, however the vast majority of the participants are associated with the UK. This should be discussed in the limitations section; the potential impact of this imbalance is worth some speculative commentary in the discussion and should be taken into account when interpreting the results.

8. Tables 1 & 2 – For the data presented in tables 1 and 2, is there a particular need to disaggregate the sample into two groups? Differences between the two groups are not discussed among the results nor are potential relationships between the n=70 subsample and their response. Thus, the two groups may not warrant detail here.

9. Throughout much of the second portion of the paper the language was quite confusing between the responses participants provided and the participants themselves. For instance, if I understand the method correctly (p. 11, lines 169-170), each participant generated two shuffled tracks and provided a different set of responses for each.

a. On this point, did the participants complete a full questionnaire for one track and then begin anew for the second track? Or did they generate the two tracks and then provide two responses to each survey item?

b. P. 14, line 252 – Beginning around this point, the data refer to “participants” (for example, here it reads “50% of people responded negatively”). But the unit of analysis is responses to tracks, correct? Presuming that 50% of people did not respond negatively to both tracks, how many negative responses were there?

c. P. 15, line 296 - Again confusing. Each participant answered four open questions twice!

d. P. 27, line 561 – “13.6% of participants”…again confusing in its reference to participants rather than responses.

e. Overall, please reconcile language like this throughout.

10. P. 13 – The section on coding and analysis was confusing.

a. Line 222 - Were codes chosen from among those created? Were the initial codes created before this step? What do you mean by “initial review of the data”?

b. Line 227- “had more codes”; Do you mean “assigned more codes”? Do these refer to the L3 codes?

c. Line 228 - Do you mean that 66% of Coder 3’s codes were in agreement with those of Coder 2? Does this account for missing codes? In other words, did the two coders assign different codes in the remaining 34% of the cases IN ADDITION TO the number of cases in which Coder 3 did not assign a code. The level of agreement is really rather low as it is, and would be considerably (arguably unacceptably) lower if “missing” codes are over and above this figure.

d. Line 230 – “memos were written to explicate”; I understand the reference to memos, but how were they used to accomplish this purpose?

11. P. 14, Lines 243-244 - This is confusing. Each participant wrote about two tracks, correct? Following from this, what % of the track-skippers were repeat offenders?

12. P. 14, Line 249 - Again, these numbers seem to be referring to individuals when each set of track data represents 1/2 of each participant’s contribution. (Also, please indicate if and how often you included responses from participants who only evaluated one track.)

13. P. 15, line 270 -- It would be useful to remind readers here that these data only refer to the responses of 70 participants.

14. P. 24, line 471 -- It should be pointed out that this finding emerged from a smaller subset of participants. Given this significant limitation, I am wondering whether this finding should occupy such a prominent placement among the results.

15. There are several uses of the term “methodology” throughout the discussion. I think of this term as referencing the collective set of methods used in a given discipline or research tradition. Within the scope of a single study, might the more basic “method” be more accurate?

16. P. 26, line 521-522 -- Might this be a reflection of the shuffle aspect? This may be particularly so in light of self-selection as the top listening approach.

17. P. 27, lines 563-564 -- This might be worth some extra emphasis given the high percentage of participants who used Spotify.

18. Figure 2 -- Does the rank of this response call into question the completeness of the item? Were one or more vital responses left out? Do your data provide evidence of what these might include?

Reviewer #2: This is a nicely done study which builds on the work of others (particularly the fourth author) and makes a modest but worthwhile contribution to the literature. The "shuffle play" method itself is interesting and worthwhile, but in connection with the results reported herein shows a limitation to the study: the question of "why did I download this track?" gets answered in terms of artist/genre, but clearly that broad determination does not indicate how much a track would be preferred or meaningful to the listener; there's a two-stage preference sort here, with artist/genre as a first, quite rough cut, and actual listening choice/preference/response a the second, more fine-grained filter. This points out the relatively indiscriminate nature of seeking out and using online information of any kind, not limited to music.

In any qualitative study it's hard to know how to present the results to be both brief enough for publication and deep enough to be informative and to dispel questions of bias, intentional or unintentional, in how the results are being reported. I'd personally like more examples of responses, even if only in the most oft-used categories (musical characteristics, change of feelings, etc).

It would also be really interesting to get more on the _negative_ responses and, if possible, on what tracks were skipped! This method could be quire illuminating relative to the normal "preferred" and "self-chosen" music methods but the 'contrary' side--pieces which were disliked or skipped--needs to be fleshed out more for this to be the case.

Reviewer #3: I agree that this is a novel method that provides some new insights -- in particular, insights regarding the importance of associations of tracks to information outside of the music itself. See however comments below regarding a potential distortion with regard to intrretation of this finding

Regarding the reporting of data -- I found it curious that the authors did not take advantage of the opportunity to describe differences between musicians and non-musicians -- this data appears to be available

Lines 235-236. Using Tableau, I wasn't sure whether the modifications I was making to the filters were having an impact on the visualized data because of the proportion of each subpopulation in the total sample or whether it reflected the unique views of the population remaining after filtering. I wonder if using proportions would be more informative.

Lines 254-255, Possible missing details -- change to? "some of the participants who responded NO then went on to describe ..."

Lines 313-315. for me at least, "relationship with a track" does not connote anything about emotion, regardless of what the piece is -- this is about my history with the band, song, or my associations with it -- hence, I don't see this contradicting the quantitative ranked data regarding "why I choose to listen". This interpretation seems like a bit of a distortion. I believe this point appears in different places in the manuscript --- abstract and discussion -- should be reconciled throughout

Lines 455-458. the authors note that the ability to get insights about music that listeners do not like is a unique opportunity afforded by the shuffle methodology. but this is hardly the most effective means of getting this info - - they could have asked people to name and defend tracks they'll especially dislike. The authors should consider emphasizing that this is more of a "fringe" benefit of the method rather than something it is ideally designed to address

Reviewer #4: "Do the Shuffle: Using shuffled play to explore reasons for music listening" is an exciting title that piqued my curiosity. A laudable contribution to research in music on why people download and listen to music, and which aspects of music listening experiences are people prioritize while listening to music on their devices.

I am surprised to see the names of the authors on the page, making me question the blindness of this review.

The abstract is clear and answers the purpose of the study. I am curious about the tact that people download music because they like it – Seems to highlight the emotion of liking. Earlier, the authors state that participants use "sound and musical features" to characterize music and associate the music with their context "over ... emotional responses". "Therefore, how their prioritize music seems to contradict why the music is downloaded in the first place — an interesting dilemma.

The introduction: A well-posited argument, the authors emphasize the importance of self-chosen music for research, music fit and context, functions of music listening, and music preference. They address the gaps in previous studies – music is preselected for participants — the importance of self-chosen music in the new technological era. Arguing that listeners today "carry their music with them," the authors use Shuffle Play an app that allows for the flexibility of listening choices, to collect data. The App is distinct because it places ownership in the hands of the listener while also allowing for a chance. In their introduction, the authors defend their arguments answering why they chose to research self-directed listening in the first place.

Aims and Questions: Aim to use a "mixed methods approach" to answering two questions: first, why people download and listen to music, and second, which aspects of the music listening experiences are prioritized by people listening to music on their devices.

Methods: Design: Even though the researchers tell us that they are using a mixed-methods approach, we do not know what kind of mixed methods approach. In mixed-methods, researchers collect both quantitative and qualitative data. Therefore, the plan may have combined methods, but the design is either, for example, Explanatory, sequential, convergent, or complex with embedded core designs. I encourage the researchers to read: Creswell, J. W., & Creswell, J. D. (2018). Research design: Qualitative, quantitative, and mixed methods approaches (5th ed.). Thousand Oaks, CA: Sage. They are using only a survey questionnaire for this purely descriptive study. Are the authors suggesting that Shuffle play is the qualitative part of their data collection?

Participants: The authors have a robust sample size (n.322). Excellent. It will be interesting to find out why 2/3rds are female, not sure how the authors will discuss this later. Interesting that they came from different countries. How was the sample accessed? Were they invited to participate? From a conference. Not sure about the representative sample argument – if we have single-digit representations from some countries. Using snowballing techniques from authors networks (See Procedure) does not make this representative.

Shuffle play: This has loads of potential for future research. I am not sure if I can call this qualitative data collection, primarily as a questionnaire, is also used to ask descriptive questions.

Results: Associations, as the most influential type of response then, describing characteristics of a soundtrack and finally evaluation of the participant relationship, or familiarity with the music. How participants responded to music-related directly to their emotions – that is, they liked it.

My final thoughts: While this is a very promising study, it might be best for the authors to position it as a descriptive study using two forms of data collections- one through direct survey and other through online survey within a listening app. More work is needed to support the explanation of your methods.

6. PLOS authors have the option to publish the peer review history of their article (what does this mean?). If published, this will include your full peer review and any attached files.

Reviewer #1: Yes: Steven J. Morrison

Reviewer #2: No

Reviewer #3: No

Reviewer #4: No

---

## [Author Response · Author response to Decision Letter 0]

10 Dec 2019

We would like to thank all reviewers for their thoughtful and helpful comments. Below we detail our responses to each reviewers’ comments (in italics) as well as highlight any substantial changes to the manuscript, both by including new text in this document (with page and line numbers) and providing its position in our revised document (attached). 

Editor: 

In particular, the authors need to specify the mixed methods design of this study and integrate the findings of the qualitative and quantitative components (narratively and via joint display). The authors should also clearly elucidate the novel contribution of this work.

Thank you for your comments. We have changed the description of the method, as per the suggestions made by the reviewers, to be a descriptive study using mixed response type as opposed to a mixed methods approach. This has been changed throughout the manuscript. Changes in the document have been made to reflect this: 

With this method, along with collecting mixed response type data, we build on and go beyond the more typical foci in the literature on music and emotion, music in everyday life and music and technology 

(Aims and research questions: Page 7, line 140-141)

For this descriptive study, we collect responses from a mix of question response types, including closed and open questions.

 (Design: Page 7, line 152-153)

By analysing responses from both open and closed questions a better understanding of the potential reasons this option was chosen so often can be achieved. 

(Discussion: Page 26, lines 524-526)

In terms of the integration of the different types of responses, we have added filters within the visualisations that enable the exploration of how these types of responses relate. We have also pointed to this within the manuscript. 

While this capability is not extensively used in this paper, the interactivity provided by the filters allows the exploration of relationships of the different codes with certain demographic factors and responses to the closed questions.

(Data Visualisations, Page 15, line 267)

We have highlighted the main contribution of the study within the conclusion of the manuscript. 

While the specific findings are not revolutionary, the new method and the mix of response type data used allows for the illumination of the diversity and subtlety of personal listening experiences.

(Conclusion, Page 32, lines 664-666)

Reviewer 1: 

In this paper the authors examined factors that lead individuals to download and/or listen to specific music selections. Rather than responding to pre-selected music examples or selecting their own music examples, participants responded to two pieces that were generated by their device’s “shuffle” feature. This was an innovative design twist that allowed for a potentially richer set of data, tapping into choices that at once represented the participants’ own volition yet did not necessarily represent their strongest preferences.

Overall, I found this to be an interesting paper that nicely balanced precision necessary for survey inquiry with the more natural context-dependent reality of personal music choices. As suggested by the authors, the findings were not ground-breaking in their novelty; rather, they underscored the fluidity and diversity of the personal music listening experience.

Thank you for your kind comments

I do have several comments and suggestions for the authors. These are largely focused on aspects needing greater clarity or consistency, however, as such, are consequential for helping the reader to understand and interpret the data and interpretation presented.

1. Abstract - The title/abstract makes reference to both exploring “reasons for music listening” and the innovative aspect of the shuffle play method. Each of these issues receives different emphases throughout the paper making it somewhat unclear exactly which is the main focus of the paper. Was the main takeaway the insight into listening or the viability of the method?

From our perspective the main aim is to gain insight into music listening through this new method. As it is a new method we wanted to make sure we explained it clearly throughout the paper. While keeping the Do the Shuffle short title, we have changed the longer title to now mention the aim of investigating music listening first then the shuffled play method. 

Do the Shuffle: Exploring reasons for music listening though shuffled play

(Title: Page 2, lines 24-25)

2. Abstract – To what degree might “because they like it” subsume the various other more fine-grained motivations usually associated with music listening (e.g., mood regulation)? Is “liking” really a different category of reason or a broader umbrella term used by the participants to describe a range of listening experiences?

We agree that this is something that needs to be more carefully discussed the data collected here does not help distinguish whether this two-step process is being used. We have added a sentence in the discussion to acknowledge this. 

From this study however, it is difficult to ascertain whether “Because I like it” is an independent category or an umbrella term used that incorporates the various other more fine-grained motivations usually associated with music listening like mood regulation.

(Discussion: Page 26, lines 529-532)

3. P. 3, line 46 – “of the time” seems a bit vague. Does this refer to any time? Some specific subset of time?

This refers to participants’ waking hours, as tested through a range of experience sampling studies. The phrase has been replaced with ‘of people’s waking hours’. (Introduction, Page 4, line 65)

4. P. 5, line 112 – This is unclear. Was shuffle or travelling the largest category? Category of what?

All the participants in this study were travelling, hence ‘mobile participants’. This has been rephrased to the following to clarify the shuffle category point:

the mobile participants in Heye and Lamont's (26) study were listening via shuffle mode (the largest response category) while travelling. 

(Introduction, Page 6, line 131)

5. P. 6, line 121 – Here and throughout, I am not sure this represents a “mixed methods” approach as formally defined by sources such as Creswell (2010). The use of both closed- and open-ended questions provided two different types of data, however both were the result of a large-scale survey method (and, in fact, both gathered from the same group of people, using the same instrument, at the same time). Consider using a different term to convey use of this mixed response type.

After looking at the reference you provided we agree and have changed our method to reflect the mixed response type used in this study as opposed to a mixed methods approach. This has been changed throughout the manuscript. 

With this method, along with collecting mixed response type data, we build on and go beyond the more typical foci in the literature on music and emotion, music in everyday life and music and technology 

(Aims and research questions: Page 7, lines 140-141)

For this descriptive study, we collect responses from a mix of question response types, including closed and open questions.

 (Design: Page 7, lines 152-153)

By analysing responses from both open and closed questions a better understanding of the potential reasons this option was chosen so often can be achieved. 

(Discussion: Page 26, lines 524-526)

6. Table 1 – This breakdown of the sample is confusing as the different subsamples have not yet been explained in the report. The note is only marginally helpful because it references a survey item not yet presented in the method.

We originally included this to be as transparent as possible but see that it is more confusing then helpful, especially as there were no significant differences in the demographic information between the subsamples. Therefore, we have taken these subsamples out of Table 1 and 2 and included a note below Table 1. 

Note. Only 70 out of the 322 answered the question “Why do you usually listen to this track?” (described below) due to a glitch in the survey program. No significant differences in demographic information or musical experience (Table 2) were found between those that answered this question and those that did not. 

 (Table 1: Page 10, lines 167-170)

7. Table 1 - The diverse array of countries represented is impressive, however the vast majority of the participants are associated with the UK. This should be discussed in the limitations section; the potential impact of this imbalance is worth some speculative commentary in the discussion and should be taken into account when interpreting the results.

We recognize that this is a limitation and have made this clearer within the limitations section. 

In addition, the participants themselves, which primarily represents female, young adults, and people from or residing in the UK, cannot be assumed to generalise to all contexts and cultures 

(Limitations: Page 30, lines 621-623)

8. Tables 1 & 2 – For the data presented in tables 1 and 2, is there a particular need to disaggregate the sample into two groups? Differences between the two groups are not discussed among the results nor are potential relationships between the n=70 subsample and their response. Thus, the two groups may not warrant detail here.

Thanks again and we have followed your helpful suggestion. We have now taken out the subgroups in the tables and included this note for Table 2: 

Note. Only 70 out of the 322 answered the question “Why do you usually listen to this track?” (described below) due to a glitch in the survey program. No significant differences in demographic information (Table 1) or musical experience were found between those that answered this question and those that did not. 

 (Table 1: Page 10 , lines 167-170)

9. Throughout much of the second portion of the paper the language was quite confusing between the responses participants provided and the participants themselves. For instance, if I understand the method correctly (p. 11, lines 169-170), each participant generated two shuffled tracks and provided a different set of responses for each.

a. On this point, did the participants complete a full questionnaire for one track and then begin anew for the second track? Or did they generate the two tracks and then provide two responses to each survey item?

We have made revisions to try and clarify our methods and interpretation making sure to make clear when we are talking about the participants versus the responses the participants gave to two different tracks. 

Once they had responded to all open-ended questions for both tracks they went onto the second section of the questionnaire. 

(A New Data Collection Method: Page 12, lines 194-195)

Participants were presented with all the closed questions to be answered in relation to the first track they discussed in the previous section. They were reminded of the track title and artist in the questionnaire. They then answered all the questions about the second track they had discussed in the previous section, again being reminded of the track’s title and artist. 

(A New Data Collection Method: Page 12, lines 198-202)

An initial reading of the responses suggested that participants often repeated information when answering the different questions and also referred back to previous answers when talking about the same track. The responses to the four questions were coded as one unit in order to avoid redundant coding within one participant’s responses about the same track. The majority (97%) of participants answered these questions for two different tracks. Those that only answered the questions for one track were still included in the analysis. Each participant that answered the questions for two tracks (n = 387) had one paragraph of text for each track and each paragraph was coded separately. Differences between people who use music in their professional life and those that do not were investigated. There were no differences found in people responses to the closed questions and the themes identified from the open-ended questions.

(RQ2: Page 18, lines 342-350)

Occurrence refers to the number of times that a code was discussed in relation to any track. For example, if the code “artist” was identified in one participant’s response to their first and second track this would count as two occurrences coded as artist. In total there were 4,182 code occurrences. See Fig 3 for a breakdown of the code hierarchy.

(RQ2: Page 19, lines 355-357)

b. P. 14, line 252 – Beginning around this point, the data refer to “participants” (for example, here it reads “50% of people responded negatively”). But the unit of analysis is responses to tracks, correct? Presuming that 50% of people did not respond negatively to both tracks, how many negative responses were there?

We have changed this to now be clearer.

For the question, “Do you have a relationship with this track? Whether it's 'Yes' or 'No' please explain”, participants responded that they did not have a relationship with a large proportion (about 50%) of the 784 total tracks discussed. 

(Survey Experience: Page 16, lines 286-288)

c. P. 15, line 296 - Again confusing. Each participant answered four open questions twice!

Thanks again. This has been made clearer in our wording to reflect that the unit of analysis is track rather than participant. 

784 tracks were included in the overall analysis. The majority of the tracks (91%, n = 719) were rated as enjoyable (enjoyed (n =336, 43%) or really enjoyable (n =383, 49%)). Few tracks (9%, n = 67) were not rated by the participant as enjoyable (1% really did not enjoy (n = 7), 2 % did not enjoy (n =14) and 6% neither enjoyed/not enjoy (n =44). Ten tracks were not rated for enjoyment. 

(RQ1: Page 17, lines 322-326)

d. P. 27, line 561 – “13.6% of participants”…again confusing in its reference to participants rather than responses.

This was made clearer with the numbers made more explicit. 

Setting aside the process of the shuffle algorithm embedded in Spotify and other music listening applications, 44 participants, out of the 322 that answered this question, skipped the track which was chosen through the shuffle method for either one or both of the tracks (of these only five skipped the app-chosen track for both responses).

(Limitations, Page 30, lines 614-618)

e. Overall, please reconcile language like this throughout.

This has now been amended throughout. 

10. P. 13 – The section on coding and analysis was confusing.

a. Line 222 - Were codes chosen from among those created? Were the initial codes created before this step? What do you mean by “initial review of the data”?

b. Line 227- “had more codes”; Do you mean “assigned more codes”? Do these refer to the L3 codes?

c. Line 228 - Do you mean that 66% of Coder 3’s codes were in agreement with those of Coder 2? Does this account for missing codes? In other words, did the two coders assign different codes in the remaining 34% of the cases IN ADDITION TO the number of cases in which Coder 3 did not assign a code. The level of agreement is really rather low as it is, and would be considerably (arguably unacceptably) lower if “missing” codes are over and above this figure.

d. Line 230 – “memos were written to explicate”; I understand the reference to memos, but how were they used to accomplish this purpose?

We have now rewritten this whole section to make our analysis clearer. Overall, we made the language more succinct and free of unnecessary jargon. We have also addressed all the suggestions made in a-d within these changes highlighted below. 

Following Braun and Clarke (34), thematic analysis (34) was used to identify categories and themes that represented the data. Overall, categories were identified which became the first codes (level 3, L3). These codes were then grouped into themes (level 2, L2). Finally, themes were merged to develop higher-level themes (level 1, L1). 

After reviewing a subset of the data (72 responses in total), two different coders (Coder 1 and Coder 2) independently identified categories to represent key elements of the data by creating level 3 (L3) codes. Both coders then compared their codes and developed larger themes (L2) to be developed into a codebook. (See the supplementary information (S2 File) for the full codebook with definitions and examples). A new coder (Coder 3) used the codebook to code the entire data set. To verify that the final coder had interpreted the codes as intended, the codes assigned by Coder 3 were compared to the original codes assigned by Coder 2. Coder 2 assigned more codes (L3 and L2) overall (n = 436) than Coder 3 (n = 364). 66% of Coder 3’s codes were in agreement with Coder 2’s codes. This calculation included the times when Coder 3 did not assign a code. Following this, the research team discussed any discrepancies between the codes and the final coding of all the data was then finalized. Following this, higher-level themes (L1) were developed in preparation for writing this paper.

 (Analysis: Page 14, lines 244-259)

11. P. 14, Lines 243-244 - This is confusing. Each participant wrote about two tracks, correct? Following from this, what % of the track-skippers were repeat offenders?

We have now made this clearer by clearly stating the % of participants who skipped either one or both. 

278 out of 322 participants (86%) stated they had not skipped any tracks (after clicking shuffle) before writing about the tracks. 12% skipped one track, either after the first or second time they shuffled their device, while only 2% skipped for both tracks. 

(Survey Experience, Page 15, lines 269-272)

12. P. 14, Line 249 - Again, these numbers seem to be referring to individuals when each set of track data represents 1/2 of each participant’s contribution. (Also, please indicate if and how often you included responses from participants who only evaluated one track.)

We have made this clearer in this section by stating that the enjoyment levels were calculated by track not participant. We also explained in more detail how many participants answered the open-ended questions about two tracks as opposed to one. 

The average enjoyment level was 4.34 (0.79) out of 5 for the tracks discussed by participants that did not skip a track, while the average for the tracks discussed by those that skipped one or more tracks was 4.33 (0.66). Further investigation showed that there were no relevant differences in the distribution of assigned codes, as illustrated by the skipped track filter in the interactive visualisation. Therefore, responses by participants who had skipped tracks before responding were combined with those by participants who had not skipped tracks in the analysis. The majority (n = 387) of 397 participants answered all the open-ended questions for two different tracks. The responses from the participants that only answered the questions for one track (n = 10) were also included in the analysis. 

(Survey Experience, Page 15, lines 272-281) 

13. P. 15, line 270 -- It would be useful to remind readers here that these data only refer to the responses of 70 participants.

This is a helpful suggestion and a small clarification has been included. 

While this data was only collected from 70 participants due to a glitch in the online survey, these top five most relevant options make up 60% of the total options ranked by those 70 participants (Fig 2). 

(RQ1: Page 17, lines 308-310) 

14. P. 24, line 471 -- It should be pointed out that this finding emerged from a smaller subset of participants. Given this significant limitation, I am wondering whether this finding should occupy such a prominent placement among the results.

We have pointed this out now within the results (above) and we acknowledge this is a limitation in the limitations section. Having said that, the original paper on which the question was based included only 25 participants. We therefore feel that though this is a small subset of participants compared to the rest of the study, it is still a useful sample and can provide a contribution to our understanding of reasons why people listen to music. 

Due to a glitch in the online survey, only 70 participants ranked the reasons why they listen to 140 tracks. The original paper by Greasley and Lamont (1), from which this question was based on, included 25 participants. Therefore, the interpretation of these current results, though taken from a smaller subgroup of our sample, is significant in comparison with other studies in this field 

(Limitations: Page 30, lines 627-631)

15. There are several uses of the term “methodology” throughout the discussion. I think of this term as referencing the collective set of methods used in a given discipline or research tradition. Within the scope of a single study, might the more basic “method” be more accurate?

Methodology has now been changed to method throughout the discussion. 

16. P. 26, line 521-522 -- Might this be a reflection of the shuffle aspect? This may be particularly so in light of self-selection as the top listening approach.

We have made this clear within the section by including shuffled play as an important aspect which contributes to the specific results interpreted within the study.

 This study shows that this is not an aspect of the music listening experience that people consistently prioritize in a shuffled play situation.

(Discussion, Page 28, lines 576-578)

We have also added a sentence in the limitation section to make it clear we do understand the impact shuffled listening has on people’s responses. 

We also consider that the results of this study are a reflection of the specific Shuffled Play method utilized in this study.

(Limitations: Page 30, lines 610-611)

17. P. 27, lines 563-564 -- This might be worth some extra emphasis given the high percentage of participants who used Spotify.

We have added another sentence to further explain how this limitation affects the interpretation of the results. 

In this case our method limited the tracks selected to no longer represent participants’ full library but rather a selection of their library or playlist. 

(Limitations, Page 30, lines 620-621)

18. Figure 2 -- Does the rank of this response call into question the completeness of the item? Were one or more vital responses left out? Do your data provide evidence of what these might include?

This is an interesting point but we have no way of knowing what responses might have been left out. Our rational for doing a top five ranking was to force people to think about the responses which represented the most salient or important reasons to help us hone in on the most important possibilities. 

Reviewer #2: 

This is a nicely done study which builds on the work of others (particularly the fourth author) and makes a modest but worthwhile contribution to the literature. The "shuffle play" method itself is interesting and worthwhile, but in connection with the results reported herein shows a limitation to the study: the question of "why did I download this track?" gets answered in terms of artist/genre, but clearly that broad determination does not indicate how much a track would be preferred or meaningful to the listener; there's a two-stage preference sort here, with artist/genre as a first, quite rough cut, and actual listening choice/preference/response a the second, more fine-grained filter. This points out the relatively indiscriminate nature of seeking out and using online information of any kind, not limited to music.

Thank you for this interesting point. We can understand how that can be one interpretation of the results and a potential limitation of our method. We cannot say, based on our collected data, if this filtering is being done by our participants or if they are even cognizant of making these types of stepped decisions. We do however think this is an important possibility to consider and therefore have discussed this further in the discussion section of the paper.

It may also be the case that people have a two-step process when thinking about why they download music that might be indicative of how people search and find music online. First decisions might be made based on genre or artist preference and following this judgment, preference around the sound and the music itself may be made. 

(Discussion, Page 28, lines 568-572)

In any qualitative study it's hard to know how to present the results to be both brief enough for publication and deep enough to be informative and to dispel questions of bias, intentional or unintentional, in how the results are being reported. I'd personally like more examples of responses, even if only in the most oft-used categories (musical characteristics, change of feelings, etc).

We will include the full data set with the responses and assigned codes in the supplementary material. We also have added a tab to the visualization called “Participants responses" which includes the examples of responses. 

It would also be really interesting to get more on the _negative_ responses and, if possible, on what tracks were skipped! This method could be quire illuminating relative to the normal "preferred" and "self-chosen" music methods but the 'contrary' side--pieces which were disliked or skipped--needs to be fleshed out more for this to be the case.

We agree that this would be interesting but unfortunately we did not collect this information. We discuss what we can about the negative responses to the tracks but few tracks were disliked. We have added sentences in the future directions section about the possibility to study this in future research. 

While this study included responses to some music that participants disliked, the participants liked the majority of tracks they discussed and no data was collected on the tracks that were skipped. A modification of this method could usefully intervene at the moment of skipping tracks to understand why people might dislike a track or in what contexts and why people might skip a track when listening to music using Shuffled Play. 

(Implications for future research: Page 31, lines 651-655)

Reviewer #3 

I agree that this is a novel method that provides some new insights -- in particular, insights regarding the importance of associations of tracks to information outside of the music itself. See however comments below regarding a potential distortion with regard to interpretation of this finding

Regarding the reporting of data -- I found it curious that the authors did not take advantage of the opportunity to describe differences between musicians and non-musicians -- this data appears to be available

We did investigate this but found no differences between musicians and non-musicians. We added a filter to the visualization so readers could explore this for themselves. 

Differences between people who use music in their professional life and those that do not were investigated. There were no differences found in people responses to the closed questions and the themes identified from the open-ended questions. 

(RQ2: Page 18, lines 347-350)

Lines 235-236. Using Tableau, I wasn't sure whether the modifications I was making to the filters were having an impact on the visualized data because of the proportion of each subpopulation in the total sample or whether it reflected the unique views of the population remaining after filtering. I wonder if using proportions would be more informative.

We are unsure to which particular visualisation you are referring, but in all cases, the filtering is configured by default (though it can be changed with a right click on it) to remove the individuals/responses that are not ticked. We agree that by using proportions it would be easier to see how relative values change between classes when some individuals are removed. We have made this change in the labels, while still maintaining the counts in the axis. 

Lines 254-255, Possible missing details -- change to? "some of the participants who responded NO then went on to describe ..."

Thank you this has now been changed. 

Interestingly, some of those participants who answered “no” describe aspects that fit with our understanding of the term ‘relationship’ within their responses (e.g. enjoyment, used for a specific purpose, associations etc.).

 (Survey Experience: Page 16, lines 288-289)

Lines 313-315. for me at least, "relationship with a track" does not connote anything about emotion, regardless of what the piece is -- this is about my history with the band, song, or my associations with it -- hence, I don't see this contradicting the quantitative ranked data regarding "why I choose to listen". This interpretation seems like a bit of a distortion. I believe this point appears in different places in the manuscript --- abstract and discussion -- should be reconciled throughout.

Thank you for this comment. We have tried to make our point clearer within the manuscript. When preparing the survey we had the same assumption as you. However, when looking at the data, we found that most participants did not share this same assumption. For example, if “relationship with this track” is not about emotion and rather about associations then most people who see this questions would answer Yes and then describe how they always liked the band or remember hearing it with their friends. However this was not the case. About half the people answered with responses like: “No I don’t but I have always liked the band and I remember hearing it with my friends.” 

For the question, “Do you have a relationship with this track? Whether it's 'Yes' or 'No' please explain”, participants responded that they did not have a relationship with a large proportion (about 50%) of the 784 total tracks discussed. Interestingly, some of those participants who answered “no” describe aspects that fit with our understanding of the term ‘relationship’ within their responses (e.g. enjoyment, used for a specific purpose, associations etc.). With some of the participants responding “no”, our findings suggest that our understanding of the word ‘relationship’ in the context of music listening may not be representative of how a substantial group of our respondents think about the music they listen to, associating this phrase with an emotional response rather than an associational one. 

 (Survey Experience: Page 16, lines 286-294)

Lines 455-458. the authors note that the ability to get insights about music that listeners do not like is a unique opportunity afforded by the shuffle methodology. but this is hardly the most effective means of getting this info - - they could have asked people to name and defend tracks they'll especially dislike. The authors should consider emphasizing that this is more of a "fringe" benefit of the method rather than something it is ideally designed to address

We agree there are better ways to understand people’s responses to music they do not like and agree this is a fringe benefit of this method. We have been more explicit that the method allows for a variety in responses which may include some negative ones. 

Finally, by using the Shuffled Play method, we were able to elicit a variety in people’s responses. This included some tracks participants did not enjoy or did not like specific aspects of. This helps highlight the nuanced way in which people talk about music. 

(Results Summary: Page 25, lines 503-506)

Reviewer #4: 

"Do the Shuffle: Using shuffled play to explore reasons for music listening" is an exciting title that piqued my curiosity. A laudable contribution to research in music on why people download and listen to music, and which aspects of music listening experiences are people prioritize while listening to music on their devices.

The abstract is clear and answers the purpose of the study. I am curious about the tact that people download music because they like it – Seems to highlight the emotion of liking. Earlier, the authors state that participants use "sound and musical features" to characterize music and associate the music with their context "over ... emotional responses". "Therefore, how their prioritize music seems to contradict why the music is downloaded in the first place — an interesting dilemma.

Thank you for this interesting observation. We interpret your point here as being based on the assumption that liking music is an emotional response in itself. In this paper we do not start from this assumption, rather liking is taken to be an aesthetic judgment rather than an emotional response. See also, for example, Juslin’s framework (2013): “the present framework postulates that an aesthetic judgment can lead to both liking (i.e., preference) and emotion. Liking (or disliking) is a mandatory outcome, whereas emotion is a possible additional outcome.” 

The introduction: A well-posited argument, the authors emphasize the importance of self-chosen music for research, music fit and context, functions of music listening, and music preference. They address the gaps in previous studies – music is preselected for participants — the importance of self-chosen music in the new technological era. Arguing that listeners today "carry their music with them," the authors use Shuffle Play an app that allows for the flexibility of listening choices, to collect data. The App is distinct because it places ownership in the hands of the listener while also allowing for a chance. In their introduction, the authors defend their arguments answering why they chose to research self-directed listening in the first place.

Aims and Questions: Aim to use a "mixed methods approach" to answering two questions: first, why people download and listen to music, and second, which aspects of the music listening experiences are prioritized by people listening to music on their devices.

Methods: Design: Even though the researchers tell us that they are using a mixed-methods approach, we do not know what kind of mixed methods approach. In mixed-methods, researchers collect both quantitative and qualitative data. Therefore, the plan may have combined methods, but the design is either, for example, Explanatory, sequential, convergent, or complex with embedded core designs. I encourage the researchers to read: Creswell, J. W., & Creswell, J. D. (2018). Research design: Qualitative, quantitative, and mixed methods approaches (5th ed.). Thousand Oaks, CA: Sage. They are using only a survey questionnaire for this purely descriptive study. Are the authors suggesting that Shuffle play is the qualitative part of their data collection?

Thank you for this helpful reference. You comment, along with that of reviewer 1 has helped us better describe our method. We have adjusted the manuscript to talk about the study as using mixed response type data.

With this method, along with collecting mixed response type data, we build on and go beyond the more typical foci in the literature on music and emotion, music in everyday life and music and technology. 

(Aims and research questions: Page 7, lines 140-141)

For this descriptive study, we collect responses from a mix of question response types, including closed and open questions. 

(Design: Page 7, lines 152-153)

Participants: The authors have a robust sample size (n.322). Excellent. It will be interesting to find out why 2/3rds are female, not sure how the authors will discuss this later. 

We were aware of this during recruitment and we tried to work to recruit more men, through snowball sampling. However, we found this difficult. We now discuss this as a possible limitation in the limitation section. While this is a limitation it is not uncommon to have more women respondents for surveys (e.g. Curtin, R., Presser, S., & Singer, E. (2000). The effects of response rate changes on the index of consumer sentiment. Public Opinion Quarterly 64: 413–428.)

In addition, the participants themselves, which primarily represents female, young adults, and people from or residing in the UK, cannot be assumed to generalise to all contexts and cultures. 

(Limitations Page 30, lines 621-623)

Interesting that they came from different countries. How was the sample accessed? Were they invited to participate? From a conference. Not sure about the representative sample argument – if we have single-digit representations from some countries. Using snowballing techniques from authors networks (See Procedure) does not make this representative.

We have tried to better explain that the initial networks invited to participate were from the authors’ personal and professional networks and were recruited via email and social media. We cannot find where we called the sample representative as this is a convenience sample. Please see our edits to the procedure section as we hope this now is clearer. 

A snowball sampling (30,31) recruitment procedure was used with direct email invitations sent to target participants from the authors' professional and social networks. 

(Procedure, Pages 13, lines 227-228)

Shuffle play: This has loads of potential for future research. I am not sure if I can call this qualitative data collection, primarily as a questionnaire, is also used to ask descriptive questions.

We agree and now have changed the methods section to better represent what was done. 

With this method, along with collecting mixed response type data, we build on and go beyond the more typical foci in the literature on music and emotion, music in everyday life and music and technology. 

(Aims and research questions: Page 7, lines 140-141)

For this descriptive study, we collect responses from a mix of question response types, including closed and open questions. 

(Design: Page 7, lines 152-153)

Results: Associations, as the most influential type of response then, describing characteristics of a soundtrack and finally evaluation of the participant relationship, or familiarity with the music. How participants responded to music-related directly to their emotions – that is, they liked it.

My final thoughts: While this is a very promising study, it might be best for the authors to position it as a descriptive study using two forms of data collections- one through direct survey and other through online survey within a listening app. More work is needed to support the explanation of your methods.

Thanks again for pointing this out. This has hopefully been addressed in our response to the previous comments.

---

## [Editor Report · Decision Letter 1]

16 Jan 2020

Do the Shuffle: Exploring reasons for music listening though shuffled play

PONE-D-19-17641R1

Dear Dr. Spiro,

We are pleased to inform you that your manuscript has been judged scientifically suitable for publication and will be formally accepted for publication once it complies with all outstanding technical requirements.

With kind regards,

Sarah E.P. Munce, PhD

Academic Editor

PLOS ONE
---

## [Editor Report · Acceptance letter]

31 Jan 2020

PONE-D-19-17641R1 

Do the Shuffle: Exploring reasons for music listening through shuffled play. 

Dear Dr. Spiro:

I am pleased to inform you that your manuscript has been deemed suitable for publication in PLOS ONE. Congratulations! Your manuscript is now with our production department. 

With kind regards,

on behalf of

Dr. Sarah E.P. Munce 

Academic Editor

PLOS ONE